# Weather Conditions and Outdoor Fall Injuries in Northwestern Russia

**DOI:** 10.3390/ijerph17176096

**Published:** 2020-08-21

**Authors:** Tatiana N. Unguryanu, Andrej M. Grjibovski, Tordis A. Trovik, Børge Ytterstad, Alexander V. Kudryavtsev

**Affiliations:** 1Department of Community Medicine, UiT The Arctic University of Norway, N-9037 Tromsø, Norway; tordis.a.trovik@uit.no (T.A.T.); boergey@online.no (B.Y.); ispha09@gmail.com (A.V.K.); 2Arkhangelsk International School of Public Health, Troitsky Ave., 51, Northern State Medical University, 163000 Arkhangelsk, Russia; andrej.grjibovski@gmail.com; 3Al-Farabi Kazakh National University, Almaty 050040, Kazakhstan; 4West Kazakhstan Marat Ospanov State Medical University, Aktobe 030019, Kazakhstan; 5Sechenov First Moscow State Medical University (Sechenov University), 119991 Moscow, Russia

**Keywords:** outdoor fall injuries, injury registry, weather conditions, Shenkursk

## Abstract

This study aimed to investigate associations between the weather conditions and the frequency of medically-treated, non-fatal accidental outdoor fall injuries (AOFIs) in a provincial region of Northwestern Russia. Data on all non-fatal AOFIs that occurred from January 2015 through June 2018 (*N* = 1125) were extracted from the population-based Shenkursk Injury Registry (SHIR). Associations between the weather conditions and AOFIs were investigated separately for the cold (15 October–14 April) and the warm (15 April–14 October) seasons. Negative binomial regression was used to investigate daily numbers of AOFIs in the cold season, while zero-inflated Poisson regression was used for the warm season. The mean daily number of AOFIs was 1.7 times higher in the cold season compared to the warm season (1.10 vs. 0.65, respectively). The most typical accident mechanism in the cold season was slipping (83%), whereas stepping wrong or stumbling over something was most common (49%) in the warm season. The highest mean daily incidence of AOFIs in the cold season (20.2 per 100,000 population) was observed on days when the ground surface was covered by compact or wet snow, air temperature ranged from −7.0 °C to −0.7 °C, and the amount of precipitation was above 0.4 mm. In the warm season, the highest mean daily incidence (7.0 per 100,000 population) was observed when the air temperature and atmospheric pressure were between 9.0 °C and 15.1 °C and 1003.6 to 1010.9 hPa, respectively. Along with local weather forecasts, broadcasting warnings about the increased risks of outdoor falls may serve as an effective AOFI prevention tool.

## 1. Background

Falls and fall-related injuries represent a serious health problem worldwide, especially in regions with a cold climate [1,2,3]. In 2015, age-standardized mortality rates from accidental falls in the Nordic countries and Russia were 1.3 times higher than those for the entire World Health Organization European Region (5.2 per 100,000) [4,5]. Falls are accountable for the largest shares of injury-related deaths and hospitalizations among older adults [6,7], and they are also associated with substantial healthcare expenditures [2,8].

In Northern geographic regions, outdoor falls occur more frequently than indoor falls. Studies carried out in Norway, Sweden, the UK, the USA, and Canada have demonstrated that between 55% and 62% of falls among middle-aged and older adults occur outdoors [2,9,10,11,12]. Streets, sidewalks, and curbs are the most common locations of outdoor falls [7,11,13]. Compared to indoor falls, outdoor falls are associated with a higher level of physical activity [11,14]. A review of prospective observational studies has shown that outdoor falls are more common in elderly people with better health-related parameters like lower body mass index, higher walking speed, and fewer diseases [15]. Moreover, the number of outdoor falls is influenced by environmental hazards, surface irregularities, weather and its seasonal variations [2,11,16,17].

The incidence of fall-related injuries is characterized by the seasonal variation in different geographic regions, including countries with cold climate (Canada, Sweden, Finland, and Norway) [3,13,17,18] and countries with warm and subtropical climate (Taiwan, Spain, Iran, and Hong Kong) [19,20,21,22]. Winter-related environmental conditions (low air temperature, snowfalls, and darkness) can cause slipperiness and increased dangerousness of sidewalks and streets [23,24]. Exposure to slippery ground surfaces, such as those covered with ice, melting ice, snow-covered ice, melting snow, and compressed snow is a common attribute of winter-time falls [16,23,24,25].

The weather conditions of the Russian North can be described as severe, with long, cold winters, sometimes heavy snowfalls, and very strong winds. However, studies investigating the association between weather conditions and outdoor fall injuries at the population level have not been carried out in this region. Knowledge of this association could be used to develop more effective prevention through the creation of risk-reducing environments, as well as through raising population awareness about the risk factors for falls [2,11]. 

The aim of the study was to investigate associations of weather conditions and the frequency of medically-treated, non-fatal accidental outdoor fall injuries (AOFIs) in a provincial region of Northwestern Russia. An accidental fall injury was defined as “inadvertently coming to rest on the ground, floor or other lower level” (International Classification of Diseases, Revision 10 [ICD-10] codes W00-19) [26]. An outdoor fall was defined as “one occurring outside a dwelling or building” [7,11,14]. 

## 2. Methods

### 2.1. Study Area

The present study was conducted in the Shenkursk District of the Arkhangelsk Region, which is located in Northern European Russia. In 2015, the Shenkursk District had a population of 13,530, which dropped to 12,610 in 2018 [27,28]. The district is largely rural, with 62% of the population living in the countryside. Due to the Northern location of the district (62°06′ N, 42°54′ E), temperatures below zero prevail from October to April [29].

The local economy is based mainly on forestry, woodworking, and agriculture. The street and walkway infrastructure is poorly developed; walkways are largely made of wood or concrete, or have a gravel surface, and street lights are placed only on major streets. Healthcare in the district is provided by one central district hospital (CDH) that has in-patient facilities and out-patient polyclinics and by rural healthcare facilities, including two out-patient clinics and 23 local units served by nurses. Approximately 80% of injuries in the Shenkursk District are treated at the CDH [30]. 

### 2.2. Study Population and Injury Data 

Data on non-fatal AOFIs were obtained from the Shenkursk Injury Registry (SHIR) for the period from 1 January 2015 through 30 June 2018 [30,31]. The SHIR gathers information on all injuries (ICD-10 codes from S00 to T78) treated at the Shenkursk CDH. Data are collected through the use of a universal injury registration form (IRF)—a two-page sheet with several sections for recording information about injured patients, including socio-demographic characteristics (sex, date of birth, address, place of work or study), information about time and place of injury, alcohol consumption in the 24 h prior to injury, use of protective equipment, and special sections for descriptions of road traffic and sports injuries. The IRF also has a mandatory section in which a free-text description of how the injury occurred is included. This section includes three supportive questions aimed to structure the description of injury circumstances: “What were you doing?”, “What went wrong?”, and “How were you injured?”. The free-text replies to these questions are transformed into categorical variables in the SHIR by trained and calibrated registrars, who define the three corresponding mechanisms (i.e., the mechanism of preceding activity, the accident mechanism, and the injury mechanism) using the appropriate coding lists. 

Injured patients who are treated in the Shenkursk CDH are asked to fill out the IRF at their first outpatient or ambulance visit, or within a few days of hospitalization. If necessary, relatives, a nurse, or a physician can help the injured patient complete the IRF. If the IRF is not filled out due to a patient’s severe condition or other reasons, injury registrars complete the form retrospectively (about 40% of cases). For that purpose, they use data from routine medical records (ambulance journal, outpatient medical card, case history) and information obtained from the attending physician. Attending physicians have to fill out the concluding part of the IRF, which includes the diagnosis with the corresponding ICD-10 code, injury severity according to the Abbreviated Injury Scale (AIS), and a record on whether the patient was hospitalized. A more detailed description of the SHIR and the IRF can be found elsewhere [30,31]. 

The SHIR variables used in this study were sex, age, injury localization by ICD-10 code, injury severity according to the AIS, hospitalization (yes, no), day (weekday or weekend) and time of the injury, the mechanism of preceding activity, the accident mechanism, and the injury mechanism. 

### 2.3. Weather Conditions

Data on weather conditions for the period from 1 January 2015 through 30 June 2018 in the Shenkursk District were obtained from the website of the Raspisaniye Pogodi Ltd., St. Petersburg, Russia [32], which includes daily archive records from the Shenkursk weather station. The weather station measures air temperature (°C), atmospheric pressure (hPa), wind speed (m/s), and relative humidity (%) eight times a day (every 3 h); amount of precipitation (mm) per 12 h is measured twice a day (at 6:00 a.m. and 6:00 p.m.; only measurements taken at 6:00 p.m. were used); and ground surface conditions are measured once a day at 6:00 a.m. (dry or moist for the warm season; no snow, covered with loose dry snow, and covered with compact or wet snow for the cold season). All of these variables were used in this study.

Assuming that associations between weather conditions and daily numbers of AOFIs can be non-linear, mean daily air temperature (°C), mean daily atmospheric pressure (hPa), mean daily wind speed (m/s), and mean daily relative humidity (%) were categorized as “low”, “medium”, or “high” for each season, using the 1st and the 2nd tertiles as cut-off values. The same categorization for each season was used for the amount of precipitation (mm); however, in the warm season, all values below the first tertile were equal to zero. For this reason, the three categories of daily amount of precipitation were labeled as “none”, “low”, and “medium/high”. 

### 2.4. Data Analysis

The associations between the weather conditions and the AOFIs were investigated separately for the cold season (15 October to 14 April) and the warm season (15 April to 14 October). Categorical characteristics of AOFIs are presented as absolute numbers and percentages. Chi-squared tests were used to compare the characteristics of AOFIs in the two seasons. 

We applied negative binomial regression for the cold season and zero-inflated Poisson regression for the warm season to model daily numbers of AOFIs, with categorized weather characteristics entered as regressors [33]. The “countfit” function in Stata was used to select the most appropriate models, based on the Akaike Information Criterion and Bayesian Information Criterion [33]. Robust standard errors were calculated for all estimates to adjust for heterogeneity in the models. All two-way interactions between weather condition variables were investigated by entering the corresponding interaction terms into multivariable models. Average percent changes (APCs) with 95% confidence intervals (CIs) were estimated by regression models to assess changes in daily numbers of AOFIs per one-unit change in each independent weather condition variable. 

Heat-maps of the daily incidence of AOFIs per 100,000 of the total population of Shenkursk District were built for each season and for combinations of the significant weather predictors. All statistical analyses were performed using STATA v. 16.1 (StataCorp LLC, 2020, College Station, TX, USA ). 

## 3. Results

The present study included 651 days of the cold season and 626 days of the warm season. The median air temperature in the cold season was −3.1 °C, and in the warm season, it was 12.2 °C (Table 1). In the warm season, 2.2% of days had a mean daily air temperature <0 °C, while this was the case for 73.1% of days in the cold season. The median atmospheric pressure and amount of precipitation were similar in the two seasons, whereas the median relative humidity was higher in the cold season. Besides, in the cold season, the ground surface was covered with loose dry snow in 64.8% of the total days; it was covered with compact or wet snow in 25.0% of the days (Table 1). 

There were 1125 non-fatal AOFIs recorded in the SHIR in the study period (Table 2). This constituted 21% of all injuries registered in the SHIR for that period and 73% of all accidental fall injuries (indoor and outdoor). The mean daily number of AOFIs in the study period was 0.88, and this number was 1.7 times higher in the cold season compared to the warm season (1.10 vs. 0.65, respectively). Correspondingly, the mean daily incidence of AOFIs in the study period was 6.7 per 100,000 population, 8.4 in the cold season and 5.0 in the warm season. The median daily incidence of AOFIs in the study period was 7.7 per 100,000 population. The distribution of AOFIs by sex, injury severity, proportion of hospitalizations, proportion of AOFIs that occurred on weekends, and by time of day was not significantly different between the two seasons (Table 2). The proportion of AOFIs among children was higher in the warm season (*p* < 0.001), and the proportion of trunk injuries was higher in the cold season (*p* = 0.023). Notably, 12.9% of adults with AOFIs in the cold period and 11.7% in the warm period reported drinking alcohol in the preceding 24 h (*p* = 0.774). 

The distribution of AOFIs by the mechanism of the preceding activity, accident mechanism, and injury mechanism was significantly different in the two seasons (*p* < 0.001) (Table 3). Walking was a more frequent mechanism of preceding activity in the cold season compared to the warm season (64.7% vs. 45.1%). Such mechanisms of preceding activity as standing and sitting, climbing up/down, and running were more frequent in the warm season. AOFI cases in the warm season who had climbing up/down and running recorded as the mechanism of preceding activity were largely children (13.9% and 27.0% respectively). The most common accident mechanism in the cold season was slipping (82.7%), while stepping wrong or stumbling over something was the most typical (48.8%) accident mechanism in the warm season. A fall on the same level was the dominating injury mechanism in both seasons, but the proportion of injuries due to falls from a height was relatively higher in the warm season (15.4% vs. 3.5%) (Table 3).

The results of univariate regression analyses showed that, in the cold season, daily numbers of AOFIs were associated with air temperature, atmospheric pressure, amount of precipitation, and a ground surface covered by snow (Table 4). Additional regression that used a binary weekend variable (Saturday and Sunday coded as “yes” and other weekdays as “no”; a proxy measure for alcohol consumption) as a regressor was not significant (*p* = 0.471). Multivariable regression for the cold season that included all weather condition variables showed that the highest daily numbers of AOFIs occurred on days when the mean daily air temperature was medium (−7.0 °C to −0.7 °C). Daily numbers of AOFIs were significantly lower on days with high mean daily air temperature (≥−0.6 °C; APC = −29.0%; *p* = 0.039). They were also lower on days with low mean daily air temperature (≤−7.1 °C; APC = −20.7%; *p* = 0.058), but the association did not reach the level of significance. Daily numbers of AOFIs were also significantly higher on days with medium/high precipitation (≥0.4 mm; APC = 24.3%; *p* = 0.015) relative to days with low precipitation (0.1 to 0.3 mm), and on days when the ground surface was covered by compact or wet snow compared to days when the ground surface had no snow (APC = 57.9%; *p* = 0.003) (Table 4). No interactions between independent variables in the cold season were observed.

In the warm season, the results of univariate regression analyses showed that daily numbers of AOFIs were associated with air temperature and atmospheric pressure (Table 5). The model that included the binary weekend variable rendered insignificant results (*p* = 0.848), as was the case in the cold season. Multivariable regression for the warm season that included all weather condition variables showed that daily numbers of AOFIs were significantly lower on days with high mean daily air temperature (≥15.2 °C; APC = −15.3%; *p* = 0.034) compared to days with medium air mean daily temperature (9.0 °C to 15.1 °C). Additionally, the daily numbers of AOFIs were lower on days with low mean daily atmospheric pressure (≤1003.5 hPa; APC = −15.7%; *p* = 0.015) relative to days with medium mean daily atmospheric pressure (1003.6 to 1010.9 hPa). The testing of two-way interactions between weather condition variables in the warm season detected modifications of the association between mean daily relative humidity and AOFIs by the daily amount of precipitation (*p* = 0.048) and by the ground surface condition (*p* = 0.045). The two corresponding interaction terms were entered into the final multivariable regression model and both sustained statistical significance. Additional stratified analysis of the associations between the mean daily relative humidity and daily AOFI numbers by the daily amount of precipitation and ground surface conditions (not presented) showed that low relative humidity had associations with smaller AOFI numbers (analysis of 56 observations; *p* < 0.001) only when the precipitation was low and soil was dry, while high relative humidity had associations with larger AOFI numbers when there was no precipitation but the soil was moist (114 observations; *p* = 0.018).

According to the heat-map of AOFI incidence, which included only the weather condition variables that reached statistical significance in multivariable regression models (Figure 1), the highest incidence of AOFIs (20.2 per 100,000 population) was observed during the cold season on days when the ground surface was covered by compact or wet snow, there was a medium mean daily air temperature (−7.0 °C to −0.7 °C), and there was medium/high precipitation (≥0.4 mm). The heat-map shows that the AOFI incidence in the warm season was substantially lower than that in the cold season. The relatively high AOFI incidence (7.0 per 100,000 population) in the warm season occurred on days with medium mean daily air temperature (9.0 °C to 15.1 °C) and medium atmospheric pressure (1003.6 to 1010.9 hPa).

## 4. Discussion

To the best of our knowledge, this is the first Russian registry-based study investigating the associations between weather conditions and medically-treated, non-fatal AOFIs in Northwestern Russia. The results demonstrate that the mean daily number of AOFIs in the cold season was 70% higher than that in the warm season. In the cold season, daily numbers of AOFIs were independently associated with air temperature, amount of precipitation, and ground surface conditions. Days that had a combination of air temperature between −7.0 °C and −0.7 °C, 2, an amount of precipitation above 0.4 mm, and a ground surface covered with compact or wet snow were described as the most “risky days” with respect to AOFIs. In the warm season, daily numbers of AOFIs were independently associated with air temperature and atmospheric pressure. The highest AOFI incidence was observed on days with an air temperature between 9.0 °C and 15.1 °C, and an atmospheric pressure between 1003.6 to 1010.9 hPa. However, the highest incidence of AOFIs observed in the warm season was substantially lower than that observed in the cold season. 

Adults of working age made up the largest proportion of cases of medically-treated non-fatal AOFIs in the Shenkursk District. However, the proportion of AOFIs in children was greater in the warm season compared to the cold season. Similarly, a study from the UK found that the number of pediatric admissions with injuries and fractures increased in the summer period [34]. This can be explained by the fact that, in the warm season, children have more spare time. They are more often outside, and thus are more likely to fall outdoors. 

The seasonal variation in AOFIs that we observed in the Shenkursk District is in accordance with studies carried out both in the Nordic countries and in countries with a warm climate. In Norway, Finland, Sweden, and Canada, the distribution of fall-related injuries and fractures varied by season and had a higher occurrence during the winter months [2,3,17,18,35,36]. For example, in three urban areas in Norway (Stavanger, Trondheim, and Harstad) the incidence rate of arm fractures among older adults was 69% higher during the colder season compared with the warmer season [17]. A study conducted in northern Sweden showed that most of the fall injuries (81%) in public outdoor environments among pedestrians 65 years and older occurred during the winter period (November to April) [2]. Even studies in Hong Kong and Taiwan demonstrated an increasing likelihood of outdoor falls in the winter months, even though the winter temperatures in these countries are well above 0 °C [19,22]. In the Nordic countries, falls occur more often in the cold season because of hazardous winter-related environmental conditions and biological factors, such as a general weakening of the body due to non-optimal vitamin D status [17,36,37,38,39]. However, the evidence about the association between vitamin D and falls or fractures is inconclusive [40]. A meta-analysis by Bolland et al. [41] has not shown a preventive effect of vitamin D on fractures or falls while Bischoff-Ferrari et al. [42] has demonstrated a reduction in the total number of fractures and falls by 14% and 12%, respectively, due to consumption of 800–1000 IU vitamin D daily.

Air temperature is a well-known meteorological risk factor for outdoor falls. Many studies have shown an association between below-zero temperatures and a higher incidence of fall-related fractures [2,18,25,35]. In our study, the highest risks of AOFIs were observed on days with an air temperature between −7.0 °C and −0.7 °C. These may be considered relatively comfortable winter temperatures in which people are more likely to participate in outdoor activities compared to colder days, as well as compared to days when temperatures rise above zero and it becomes wet. In addition, De Koning et al. [43] described that the minimum friction coefficient occurs when the ice surface temperature is between −6 °C and −9 °C, which is that the worst outdoor temperature interval for walking outside, as that is when slip and fall accidents are most likely [44]. Contrary to that, a study from Finland described that the incidence of outdoor falls among elderly people was 3.4 times higher when the temperature was below −20 °C than when it was between −10 °C and 0 °C [35]. That means that associations between AOFIs and weather conditions can vary in different settings, even if they have a similar climate. This may be due to different outdoor environments, or varying traditions of outdoor activities in the winter time. This also outlines the importance of using local injury data when planning preventive activities [45]. 

Ice and snow are well-known causes of fall-related injuries. In the present study, a ground surface covered by loose dry snow or compact/wet snow increased daily numbers of AOFIs in the cold season by 41% and 63%, respectively, compared with days when the ground surface had no snow. A study performed in Canada found a significant, positive correlation (r = 0.33–0.60; *p* < 0.001) between snow depth and the number of snowy days and hip fractures for all age groups and sexes [18]. The city of Philadelphia, Pennsylvania in the USA also recorded increases in fall-related patient visits after snow and ice storms [46]. 

In the cold season, walking was the most common mechanism of activity preceding an AOFI (65%) in the Shenkursk District, while slipping was the most common accident mechanism (83%). Similar results were obtained in Umeå, Sweden, where 85% of persons injured due to slipping on ice or snow and falling outdoors were walking before the accident [13]. The results from a Finnish study showed that the number of fractures on slippery winter days and on the days immediately following was 2.5 times higher compared to non-wintertime (April 16 to October 15) [3]. During a snow crisis in Iran, slipping was the most common injury mechanism, and the frequency of injuries on icy days was 32.4% higher than that on snowy days [21]. 

We found that “the riskiest days” in terms of AOFIs in the cold season were the days with a combination of medium air temperature, medium/high precipitation, and a ground surface covered with compact or wet snow. The importance of describing these “high-risk combinations” of weather characteristics has been outlined in other studies. For example, Lépy et al. [24] described six winter scenarios with respect to slippery conditions and identified the most common scenario of slipperiness to be a combination of relative humidity above 95%, surface temperature below 0 °C, and a surface temperature lower than the dewpoint. This type of slipperiness accounted for 50% of injuries. Morency et al. also showed that a combination of below-zero temperatures, snowfalls, and freezing rain leads to excess cases of outdoor falls [25].

Our study demonstrated that, in the warm season, a higher frequency of AOFIs was observed on days with medium air temperature and medium atmospheric pressure. Previous studies of falls have scarcely addressed the role of atmospheric pressure, and only a few reported that atmospheric pressure has an effect on the incidence of hip fractures and trauma admissions [19,20]. However, it is known that low atmospheric pressure can cause headaches, a feeling of dizziness and decreases in blood pressure among weather-sensitive people [47,48,49]. The association between dizziness and falls among middle-aged adults was found by Peeters et al. [50] during the analysis of data from population-based cohort studies in Australia and the UK. One possible explanation for the phenomenon observed in our study is that days with average air temperatures and atmospheric pressures may be the most comfortable for outdoor activities, and thus the higher incidence could be due to higher outdoor exposure. 

As the incidence of AOFIs in the Shenkursk District is the highest in the cold season and slipping is the most common accident mechanism, our study suggests that efforts to prevent AOFIs should primarily target the problem of icy ground surfaces. The common approaches are spreading anti-slip materials (sand, salt, gravel) on walkways and using slip-resistant footwear. The latter may be a feasible solution for a rural setting. Protective effects of gait-stabilizing and anti-slip devices in the winter season were demonstrated in earlier studies [51,52]. In addition to these standard approaches, it may be helpful to inform people about the increased risks when such risks are expected. For that purpose, our descriptions of “high-risk days” can supplement regular weather forecasts, thus increasing people’s awareness of the increased risks of outdoor falls on days when the prognoses fit the described high-risk combinations of weather conditions. For example, regional agencies for civil defense and emergencies in Russia inform people by short message service (SMS) about inclement weather, like strong winds and snowstorms. Based on the results of this study and local weather forecasts, SMS warnings about the higher risks of outdoor falls may also become an effective AOFI prevention tool. Such messages may also support local decision making with respect to the timely initiation of snow and ice removal and the application of anti-friction materials on walkways.

The strengths of this study are its population-based design and the geographically defined area. Given the weather and other environmental and socioeconomic contexts in the Shenkursk District are similar to those of other rural settings in Northern Russia and in other northern countries, our findings may be applicable beyond the study area. Another strength is the use of data from a registry with high coverage and representativeness of the total injuries in the district [30]. Finally, we analyzed only AOFIs, while many other studies of seasonal variation in falls did not separate outdoor and indoor fall injuries, even though they have different characteristics [3,17,18,53].

A possible limitation of our study is that the SHIR included only AOFIs that were medically treated at the CDH. Therefore, our results may be less applicable to people with mild AOFI who did not seek medical care. 

In this study, we analyzed AOFIs without stratification by sex and age, although some variation in the effect of weather conditions on the incidence of AOFIs is possible by age and sex. This study should be replicated either in a larger population or over a longer period in order to obtain a more detailed age- and sex-specific analysis.

As the study has an ecological design, the outcome variable in our analyses was the daily number of AFOIs, and a calendar day was the unit of observation. This design did not allow us to control for possible behavioral factors on an individual level, e.g., alcohol consumption [54,55]. We tried testing and saw no association between the weekend variable (Saturday and Sunday vs. weekdays) as a proxy for alcohol consumption and the number of AOFIs. Moreover, factors like alcohol consumption are unlikely to be associated with the weather conditions that we identified as being associated with AOFIs. Therefore, confounding from alcohol consumption is unlikely.

## 5. Conclusions

The AOFIs in the Shenkursk District occurred more often in the cold season than in the warm season. A combination of low air temperature, medium/high precipitation, and a ground surface covered with compact or wet snow were the attributes of days with higher risks of AOFIs in the cold season. In the warm season, the numbers of AOFIs were higher on days with medium air temperature and atmospheric pressure. Larger-scale future research is required to study the impacts of weather conditions on the frequency of AOFIs by age and sex.

## Figures and Tables

**Figure 1 ijerph-17-06096-f001:**
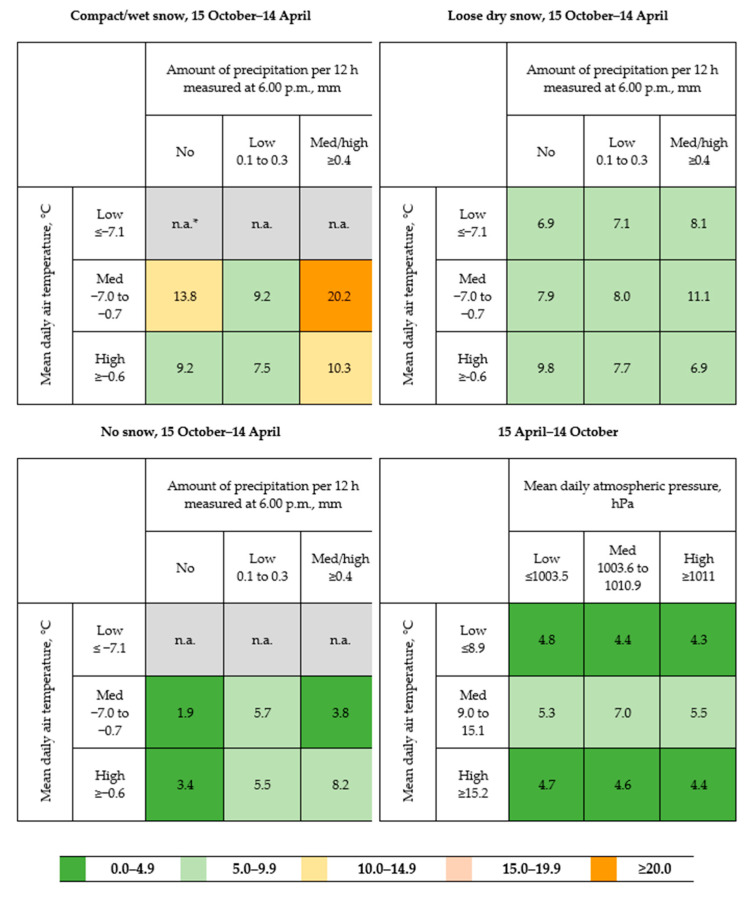
Heat-map of the mean daily incidence (per 100,000 population) of medically-treated, non-fatal accidental outdoor fall injuries in the cold season (15 October–14 April) according to ground-surface conditions, mean daily air temperature, and amount of precipitation per 12 h measured at 6.00 p.m. and in the warm season (15 April–14 October) according to mean daily air temperature and mean daily atmospheric pressure; Shenkursk District, 1 January 2015–30 June 2018. * Mean is not calculated due to lacking observations. N.a.: not applicable.

**Table 1 ijerph-17-06096-t001:** Weather conditions in the Shenkursk District by season, 1 January 2015–30 June 2018.

Weather Conditions	Cold Season 15 October–14 April (651 Days)	Warm Season 15 April–14 October (626 Days)
Mean daily air temperature, °C		
Minimum	−39.9	−9.9
1st tertile	−7.0	9.0
Median	−3.1	12.2
2th tertile	−0.6	15.2
Maximum	12.8	31.3
Mean daily atmospheric pressure, hPa		
Minimum	964.2	971.0
1st tertile	1002.2	1003.6
Median	1007.5	1007.6
2th tertile	1013.4	1011.1
Maximum	1048.3	1038.5
Mean daily wind speed, m/s		
Minimum	0.0	0.0
1st tertile	2.4	2.1
Median	2.9	2.5
2th tertile	3.5	2.8
Maximum	10.0	9.0
Mean daily relative humidity, %		
Minimum	20.0	12.0
1st tertile	82.2	65.2
Median	86.0	72.0
2th tertile	88.4	79.1
Maximum	99.0	100.0
Amount of precipitation per 12 h measured at 6:00 p.m., mm		
Minimum	0.0	0.0
1st tertile	0.1	0.0
Median	0.1	0.1
2th tertile	0.4	0.3
Maximum	10.0	14.0
Ground surface condition measured at 6:00 a.m., % of days		
Dry soil	0.0	53.8
Moist soil	10.1	41.1
Loose dry snow	64.8	1.9
Compact or wet snow	25.0	3.2

**Table 2 ijerph-17-06096-t002:** Socio-demographic, medical, and temporal characteristics of medically-treated, non-fatal accidental outdoor fall injuries in the Shenkursk District by season, 1 January 2015–30 June 2018.

Characteristics	Cold Season 15 October–14 April, % (*N* = 717)	Warm Season 15 April–14 October, % (*N* = 408)	*p*
Sex, male	48.1	51.7	0.246
Age group, years			<0.001
0–6	1.3	7.4	
7–17	17.6	22.5	
18–59	54.4	45.3	
60+	26.8	24.8	
Injury localization, ICD-10			0.023
S00-09: Head	9.6	10.3	
S10-39: Trunk	21.1	14.2	
S40-69: Upper extremity	40.7	43.1	
S70-99: Lower extremity	28.2	30.9	
Other	0.4	1.5	
Injury severity, AIS			0.816
1 Minor	58.0	56.1	
2 Moderate	31.4	33.1	
3 Severe, but not life-threatening	10.6	10.8	
Hospitalization, yes	10.7	13.2	0.210
Injury occurred on weekend *	29.8	29.4	0.878
Time of injury, hours			0.200
00:00–05:59	4.2	3.7	
06:00–11:59	20.8	18.5	
12:00–17:59	41.8	38.2	
18:00–23:59	26.4	29.6	
Not indicated	6.9	10.1	

* Saturday and Sunday. ICD-10: International Classification of Diseases, Revision 10; AIS: Abbreviated Injury Scale.

**Table 3 ijerph-17-06096-t003:** Mechanisms of medically-treated, non-fatal accidental outdoor fall injuries in Shenkursk District by season, 1 January 2015–30 June 2018.

Mechanisms	Cold Season 15 October–14 April, % (*N* = 717)	Warm Season 15 April–14 October, % (*N* = 408)	*p*
Mechanism of preceding activity			<0.001
Walking	64.7	45.1	
Carrying something	10.6	4.9	
Physical exercising	8.1	8.8	
Going on a stairs	7.4	5.6	
Working in a garden	5.2	9.6	
Standing and sitting	2.6	11.3	
Climbing up/down	0.8	5.9	
Running	0.6	8.8	
Accident mechanism			<0.001
Slipping	82.7	27	
Stepping wrong or stumbling over something	7.9	48.8	
Loss of balance and faintness	5.7	14.7	
Other	3.6	9.6	
Injury mechanism			<0.001
Fall on the same level	83.7	71.8	
Fall on a stairs	12.8	12.7	
Fall from a height	3.5	15.4	

**Table 4 ijerph-17-06096-t004:** Associations of weather condition variables with daily numbers of medically-treated, non-fatal outdoor accidental fall injuries in the Shenkursk District in the cold season (15 October–14 April).

Weather Conditions	N Days	N Cases	Simple Negative Binomial Regression		Multivariable Negative Binomial Regression *	
			APC, % (95% CI)	*p*	APC, % (95% CI)	*p*
Mean daily air temperature, °C						
Low (≤−7.1)	217	212	−26.7 (−46.9, −6.6)	0.009	−20.7 (−42.0, 0.7)	0.058
Med (−7.0 to −0.7)	217	277	Ref.		Ref.	
High (≥−0.6)	217	228	−20.4 (−41.6, 0.8)	0.072	−29.0 (−56.6, −1.4)	0.039
Mean daily atmospheric pressure, hPa						
Low (≤1002.1)	217	259	1.6 (−19.0, 22.1)	0.882	−1.4 (−21.4, 18.5)	0.888
Med (1002.2 to 1013.2)	217	255	Ref.		Ref.	
High (≥1013.3)	217	203	−22.8 (−43.4, −2.2)	0.03	−15.8 (−36.6, 5.0)	0.137
Mean daily wind speed, m/s						
Low (≤2.3)	233	234	−13.0 (−34.5, 8.4)	0.233	−5.1 (−26.3, 16.1)	0.638
Med (2.4 to 3.4)	208	238	Ref.		Ref.	
High (≥3.5)	210	245	1.9 (−19.1, 23.0)	0.857	−1.2 (−22.5, 20.2)	0.913
Mean daily relative humidity, %						
Low (≤82.1)	219	233	−5.6 (−26.2, 15.0)	0.596	−2.7 (−24.7, 19.2)	0.808
Med (82.2 to 88.3)	216	243	Ref.		Ref.	
High (≥88.4)	216	241	−0.8 (−21.6, 19.9)	0.938	−11.0 (−31.6, 9.6)	0.293
Amount of precipitation per 12 h measured at 6:00 p.m., mm						
No (0.0)	176	186	10.0 (−11.3, 31.3)	0.359	6.2 (−15.5, 28.0)	0.574
Low (0.1 to 0.3)	252	241	Ref.		Ref.	
Med/High (≥0.4)	223	290	30.7 (10.9, 50.6)	0.002	24.3 (4.7, 43.8)	0.015
Ground surface condition measured at 6.00 a.m.						
No snow	66	47	Ref.		Ref.	
Loose dry snow	422	453	41.0 (6.2, 75.8)	0.021	26.7 (−14.4, 67.8)	0.203
Compact/wet snow	163	217	62.6 (25.2, 100.0)	0.001	57.9 (20.1, 95.6)	0.003

* Constant = −6.1 (95% CI: −49.9, 37.6), *p* = 0.783 N: number; APC: average percent change; CI: confidence interval.

**Table 5 ijerph-17-06096-t005:** Associations of weather condition variables with daily numbers of medically-treated, non-fatal accidental outdoor fall injuries in the Shenkursk District in the warm season (15 April–14 October).

Weather Conditions	N Days	N Cases	Simple Zero-Inflated Poisson Regression		Multivariable Zero-Inflated Poisson Regression *	
			APC,% (95% CI)	*p*	APC,% (95% CI)	*p*
Mean daily air temperature, °C						
Low (≤8.9)	210	123	−3.9 (−16.5, 8.7)	0.544	−0.8 (−14.4, 12.8)	0.906
Med (9.0 to 15.1)	208	161	Ref.		Ref.	
High (≥15.2)	208	124	−14.2 (−27.1, −1.3)	0.030	−15.3 (−29.5, −1.1)	0.034
Mean daily atmospheric pressure, hPa						
Low (≤1003.5)	209	136	−13.8 (−26.6, 0.1)	0.035	−15.7 (−28.4, −3.1)	0.015
Med (1003.6 to 1010.9)	210	144	Ref.		Ref.	
High (≥1011)	207	128	−5.3 (−18.3, 7.8)	0.429	−6.0 (−19.6, 7.5)	0.382
Mean daily wind speed, m/s						
Low (≤2.0)	*218*	*141*	*1.9 (−11.0, 14.7)*	*0.779*	*0.6 (−13.3, 14.5)*	*0.935*
Med (2.1 to 2.7)	202	137	Ref.		Ref.	
High (≥2.8)	206	130	0.0 (−12.9, 13.0)	0.994	−1.4 (−14.5, 11.7)	0.834
Mean daily relative humidity, %						
Low (≤65.1)	211	136	2.9 (−10.5, 16.3)	0.676	−4.6 (−35.3, 26.0)	0.766
Med (65.2 to 79.0)	208	128	Ref.		Ref.	
High (≥79.1)	207	144	3.4 (−9.6, 16.4)	0.604	−12.6 (−68.9, 43.7)	0.661
Amount of precipitation per 12 h measured at 6:00 p.m., mm						
No (0.0)	294	195	4.6 (−10.0, 19.1)	0.535	−6.8 (−24.9, 11.4)	0.464
Low (0.1 to 0.2)	129	80	Ref.		Ref.	
Med/High (≥0.3)	203	133	2.5 (−12.9, 18.0)	0.747	−17.2 (−45.0, 10.7)	0.227
Ground surface condition measured at 6.00 a.m.						
Dry	337	233	Ref.		Ref.	
Moist	257	153	−4.9 (−16.0, 6.2)	0.389	4.7 (−11.1, 20.5)	0.561
Loose dry snow	12	8	10.0 (−17.8, 37.9)	0.480	29.8 (−3.3, 62.9)	0.077
Compact/wet snow	20	14	−3.3 (−26.2, 19.6)	0.777	24.5 (−3.6, 52.7)	0.088
Mean daily relative humidity * Amount of precipitation per 12 h					9.1 (−0.2, 18.1)	0.045
Mean relative humidity * Ground surface condition					−10.2 (−19.9, −0.5)	0.039

* constant = 54.0 (95% CI: 28.7, 79.2), *p* < 0.001. N: number; APC: average percent change; CI: confidence interval.

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
