# Peer review of "Weather Conditions and Outdoor Fall Injuries in Northwestern Russia"

_ijerph, 2020, doi:10.3390/ijerph17176096_

Round 1
Reviewer 1 Report
The aim of the presented study was to investigate the effects of weather conditions on the frequency of medically-treated, non-fatal accidental outdoor fall injuries (AOFI) in a provincial region of Northwestern Russia.
The study design is appropriate and the methods are well described. The homogeneous data basis benefits the study design (injury registry data, only one central district hospital).
Sentences like "In the Northern latitudes, falls occur more often in winter 278because of hazardous environmental conditions and biological factors, such as a general weakening of the body due to non-optimal vitamin D status [14, 31-34]" could be clearified. The effects of vitamin D on the primary prevention of falls are still being controversially discussed and could be presented less superficially (Bolland MJ, Grey A, Avenell A. Effects of vitamin D supplementation on musculoskeletal health: a systematic review, meta-analysis, and trial sequential analysis. Lancet Diabetes Endocrinol. 2018;6(11):847-858. doi:10.1016/S2213-8587(18)30265-1
Bischoff-Ferrari HA. Should vitamin D administration for fracture prevention be continued? : A discussion of recent meta-analysis findings. Sollte die Frakturprävention mit Vitamin D fortgesetzt werden? : Eine Betrachtung aktueller Metaanalysen. Z Gerontol Geriatr. 2019;52(5):428-432. doi:10.1007/s00391-019-01573-9). Confounding factors for accidental outdoor fall injuries, e.g alcohol consumption or neurological comorbidities would have been interesting and relevant.The authors should consider further references:
Peeters G, Cooper R, Tooth L, van Schoor NM, Kenny RA. A comprehensive assessment of risk factors for falls in middle-aged adults: co-ordinated analyses of cohort studies in four countries. Osteoporos Int. 2019;30(10):2099-2117. doi:10.1007/s00198-019-05034-2
Schepers P, den Brinker B, Methorst R, Helbich M. Pedestrian falls: A review of the literature and future research directions. J Safety Res. 2017;62:227-234. doi:10.1016/j.jsr.2017.06.020
James MK, Victor MC, Saghir SM, Gentile PA. Characterization of fall patients: Does age matter?. J Safety Res. 2018;64:83-92. doi:10.1016/j.jsr.2017.12.010
The quality of academic and scientific writing is fine.
Reviewer 2 Report
Dear Authors,
Your topic is vastly important to save lives and to cover the knowledge gap. As your study is the first Russian registry-based investigating the associations between weather conditions and medically-treated, non-fatal AOFIs in Northwestern Russia.
However, there are some comments that need addressing our justifications.
1- In line 167 (although the distribution of AOFIs was significantly different in the two seasons) have you examined the correlation between the aspects (preceding activity, accident mechanism, and injury mechanism)?
2- In Table 2: it seems the females have been excluded! Can you justify why? (as in line 355). Or the lines from 355-358 can be shifted to the conclusion section from future research.
3- To direct your conclusion: Have you find/notice any major categories in your sample (e.g construction labor, aviation staff, etc.).
4- A follow up paper is need to dig deeper in finding the optimal strategies and solutions as in line (331) " This can be done by properly removing ice from walkways, or by installing heated sidewalks and porches. Another approach is increasing friction on icy surfaces using anti-slip materials (sand, salt, gravel), or by using slip-resistant footwear."
Good luck in future work.
Kind regards,
Reviewer 3 Report
At least the following sentences in the paper are (almost) identical to those in other papers without quotation marks or references, and it should never be published as such. I just checked the Background section, and the list may not be complete. Before deciding whether to recommend keeping this paper in the review process for further consideration or not, I request the authors to thoroughly check and revise it first.
- “Falls are the leading cause of injury-related deaths and hospital admissions among older adults” (lines 41–42) is identical to a sentence in reference #11.
- "elderly persons who fall outdoors may perceive themselves as healthier because they actually are healthier and therefore venture outside their homes more often" (lines 50–51) is almost identical to a sentence in reference #12.
- “The seasonal variations in the incidence of fall-related injuries has been described in geographic regions with varying climatic characteristics, from cold, high-latitude countries such as Canada, Sweden, Finland, and Norway [14-17], to low-latitude countries with warm and subtropical climates” (lines 54–56) is almost identical to sentences in reference #15.
Author Response
Response to Reviewer 3
At least the following sentences in the paper are (almost) identical to those in other papers without quotation marks or references, and it should never be published as such. I just checked the Background section, and the list may not be complete. Before deciding whether to recommend keeping this paper in the review process for further consideration or not, I request the authors to thoroughly check and revise it first.
- “Falls are the leading cause of injury-related deaths and hospital admissions among older adults” (lines 41–42) is identical to a sentence in reference #11.
This sentence was rephrased: “Falls are accountable for the largest shares of injury-related deaths and hospitalizations among older adults [6, 7], and they are associated with substantial healthcare expenditures [2, 8]”. (lines 38 – 39)
- "elderly persons who fall outdoors may perceive themselves as healthier because they actually are healthier and therefore venture outside their homes more often" (lines 50–51) is almost identical to a sentence in reference #12.
We replaced this sentence by another one: "A review of prospective observational studies has shown that outdoor falls are more common in elderly people with better health-related parameters like lower body mass index, higher walking speed, and fewer diseases [15].". (lines 54 – 56)
- “The seasonal variations in the incidence of fall-related injuries has been described in geographic regions with varying climatic characteristics, from cold, high-latitude countries such as Canada, Sweden, Finland, and Norway [14-17], to low-latitude countries with warm and subtropical climates” (lines 54–56) is almost identical to sentences in reference #15.
This sentence was changed: " The incidence of fall-related injuries is characterized by the seasonal variation in different geographic regions, including counties with cold climate (Canada, Sweden, Finland, and Norway) [3, 13, 17, 18] and countries with warm and subtropical climate (Taiwan, Spain, Iran, and Hong Kong) [19-22]." (lines 62 – 65)
Several other changes were made in the text to minimize the similarities between the formulations in the manuscript with the formulations in the referred papers. All changes in the text are marked using the "Track Changes" function in Microsoft Word and are reflected in the cover letter.
Round 2
Reviewer 3 Report
The paper “Weather Conditions and Outdoor Fall Injuries in Northwestern Russia” attempts to identify weather conditions that are associated with the incidence of accidental non-fatal outdoor fall injuries (AOFIs). This is a cross-sectional ecological study, and injury data were obtained from the only hospital in the district of Shenkursk, Arkhangelsk, Russia, between January 2015 and June 2018. The authors conducted multivariable analyses separately for the cold and warm seasons and reported weather conditions that were associated with increased risk of injuries.
The findings would be useful, at least for the district. However, it might not be of interest to a broad audience because the district has a small population of around 13,000 and, as the authors argue, “associations between AOFIs and weather conditions can vary in different settings, even if they have a similar climate” (lines 281–282). Below are comments to improve the quality of the paper.
- Avoid the use of causal terms, such as effect, because this study investigated independent associations of multiple variables, including weather conditions with fall injuries and did not conduct causal analysis for specific weather conditions.
- From lines 80–82, I understood there was only one hospital in the district during the study period. Could you confirm if it is correct? Were there any other medical facilities that might have treated injury patients?
- Report atmospheric pressure in its SI Unit (Pa).
- What is the measurement duration of precipitation? Is it hourly or per 12 hours? Even if it is hourly, few people would think precipitation of 0.4 mm per hour is “substantial,” and replacing it with an appropriate adjective would avoid confusion. It might also be an option to change the cut-off value.
- Explain why you applied negative binomial regression for the cold season and zero-inflated Poisson regression for the warm season (lines 134–135).
- Are the regression models adjusted for population size? If not, consider doing so.
- Explain how the “countfit” function selected the models (lines 136–137).
- Elaborate on what interactions you examined (line 138–139). Are they selected two-ways, all two-ways, or all interactions, including those of more than two variables?
- By “multiple regression,” people usually mean multiple linear regression or ordinary least squares. If you meant there were multiple regressors in the regression models, consider replacing the term “multiple” with “multivariable.”
- Tables 4 and 5 show “annual percent change.” Why is it “annual”?
- Reconsider if the recommendations (lines 320–338) can be directly drawn from your study. For example, did the study investigate the effectiveness of “removing ice from walkways” on the incidence of AOFIs?
- Explain why you think “it is unlikely that mild AOFIs are associated with different weather variables when compared to those that require medical care” (lines 348–349).
- What is reference #54 for?
